# Auto-SPT: Automating Semantic Preserving Transformations for Code

## Abstract

Machine learning (ML) models for code clone detection determine whether two pieces of code are semantically equivalent, which in turn is a key building block for software-engineering tasks like refactoring and security tasks like vulnerability and malware detection. While these models are predominantly trained on clean, structured code datasets, real-world code often undergoes a variety of semantic-preserving transformations, including refactoring, minification, automated formatting, and compiler optimizations. To address this critical gap between training and test data, we propose Auto-SPT, a novel framework to *automatically construct synthetic-data generators* for code. Auto-SPT is designed to produce Semantic Preserving Transformations (SPTs) that alter a program's syntactic structure while preserving its functionality and is instantiated on top of Large Language Models (LLMs). In particular, we use LLMs to craft a diverse set of SPTs, generate strong implementations for these SPTs, and compose them to result into strong transformations. Our formal analysis shows that the diversity of SPTs impacts the strength of their composition. We then empirically demonstrate that Auto-SPT generates more diverse SPTs than existing approaches and these SPTs significantly drop the performance of state-of-the-art code clone detectors. Further experiments show Auto-SPT can be used to enhance code datasets for training, to produce code-clone detection models that are robust to real-world, adversarial code transformations.

## 1 Introduction

Machine learning (ML) is rapidly transforming the field of programming, enabling automated tasks such as code generation, bug detection, and vulnerability analysis. One of the most important tasks is *code clone detection*, where a model determines whether two pieces of code are semantically equivalent. This task enables code search, code clustering, code reuse detection, code quality assessment, and the identification of copyright violations, all common tasks in software development (Wei & Li, 2017; Juergens et al., 2009). Existing ML-based code clone models are trained and evaluated on datasets derived from clean and well-structured code repositories like GitHub and Stack Overflow (Alam et al., 2023; Svajlenko et al., 2014), with limited exposure to perturbed examples. However, at test time, these models might process code samples that underwent transformations because of code refactoring, minification, application of automated formatting rules, or as a result of compiler optimizations. There is a need to evaluate whether ML-based code cloning models are robust to real-world transformations of code (Zhang et al., 2023; Srikant et al., 2021).

Existing research has developed approaches to systematically transform code by applying Semantic Preserving Transformations (SPTs) to alter a program's syntactic structure while preserving its functionality (Hooda et al., 2024a; Wang et al., 2022; Le-Cong et al., 2024; Allamanis et al., 2021; Dong et al., 2024). Thus far, existing work utilized simple transformations like swapping independent code blocks, replacing `while` loops with `for` loops, switching `if` conditions, and renaming variables (See Table 5). While simple to implement,

such approaches face several shortcomings. First, by not exploring the space of possible transformations, they lack diversity in the types of transformations applied. Second, they often adopt heuristic-based implementations, making it hard to gauge the strength of the transformation applied. Third, they require significant implementation overhead to tailor and correctly apply the transformation. Because of these shortcomings, existing SPTs do not enable the realistic evaluation of the robustness of code cloning models.

In this work, we address this need through `Auto-SPT`, a framework to create and apply SPTs. First, `Auto-SPT` generates diverse semantic preserving transformations that can induce alterations to all aspects of the code, including its structure, syntax, and control. Second, `Auto-SPT` finds the implementation of each transformation that can maximally affects the performance of code cloning model. Third, `Auto-SPT` composes multiple such transformations, which as we demonstrate result in modifications more complex than those from a single transformation applied in isolation.

We instantiate this framework by leveraging Large Language Models (LLMs). LLMs demonstrate remarkable capabilities in coding tasks such as generating and editing code. We use them to automatically design and implement new SPTs. First, we use LLMs to synthetically craft a diverse set of SPTs. Then we utilize LLM's code generation capability combined with execution feedback (similar to Zheng et al. (2024)) to generate strong implementations for each of the designed SPTs. Finally, we demonstrate that a composition of these diverse SPTs significantly increases the strength of the resultant transformations and the robustness of clone-detection models trained on thus transformed code.

We formally analyze how the diversity of SPTs impacts their composition and propose a diversity metric to quantify it. We then empirically demonstrate that `Auto-SPT` generates stronger implementations for existing SPTs and more diverse SPTs as compared to existing approaches. Finally, we show that composing SPTs generated by `Auto-SPT` results in significantly stronger transformations. For example, our evaluation using code samples from the CodeContests dataset (Li et al., 2022) over a fully finetuned CodeBERT model shows that the average distance (where clones have a distance of 0 and non-clones have 1) between original code samples and transformed samples (semantically equivalent) from `Auto-SPT` is $0.947$ compared to $0.243$ from prior work. Even for state of the art embedding models like EmbeddingGemma (Gemma-Team et al., 2025), `Auto-SPT` is able to find transformed samples with average distance $> 0.8$.

## 2 RELATED WORK

**Clone Detection.** Code clone detection is a critical area in software engineering, aiming to identify similar code fragments within or across software systems, useful for maintenance, refactoring, bug fixing, and ensuring software quality (Shan et al. (2023)). There are different types of code clones: Type-1 are identical clones, modulo formatting and comments; Type-2 are lexical clones, where code literals and variable names vary; Type-3 are syntactic clones, where code statements are added, removed, or modified; and Type-4 are semantic clones, where functionality is preserved, but not the syntactic structure. Type-4 clones, the ones we target in this work, are the most challenging type to detect, requiring understanding the underlying logic. Early work in this space focused on Types 1, 2, and 3 through specially designed representations of the source code and corresponding matching algorithms (Kamiya et al., 2002). More recent approaches have used similarity of embeddings obtained from neural models (e.g., Alon et al. (2019b)'s code2vec, Alon et al. (2019a)'s code2seq, Feng et al. (2020b)'s CodeBERT). Despite this recent progress, the detection of Type-4 clones accurately and scalably remains an open problem, as observed through benchmarks such as GPTCloneBench (Alam et al. (2023)), where the recall of code-clone detectors is quite low for real-world code samples.

**Semantic Preserving Transformations (SPTs).** SPTs are code alterations that modify a program's syntactic or structural representation but maintain its original functionality. These transformations enable several tasks in software engineering, including software testing, robustness evaluation, and data augmentation for

training code-understanding models. Prior work has explored various semantic-preserving code transformations, notably variable renaming, conditional restructuring, loop transformations such as for/while loop conversions, swapping if/else blocks, statement reordering, and loop unrolling (Allamanis et al. (2021); Rabin et al. (2021); Bui et al. (2021); Henkel et al. (2022); Chakraborty et al. (2022); Yang et al. (2022); Zhang et al. (2023); Wang et al. (2022); Hooda et al. (2024a); Dong et al. (2024)). Variable renaming is widely studied across these works, whereas more sophisticated transformations, such as loop unrolling and conditional restructuring, have received relatively less attention. Past work has also utilized SPTs to search for adversarial programs against tasks such as program summarization (Srikant et al. (2021); Henkel et al. (2022)) and clone detection (Zhang et al. (2023)). In this work, we characterize how the set of available SPTs impacts the program search, and we provide an automated framework to generate new SPTs.

**Large Code Models.** LLMs have become powerful tools for various software engineering tasks, including code generation, completion, bug detection, and automated refactoring. This ability has unlocked new applications where the model generates code as an intermediate representation to solve downstream tasks instead of directly outputting the final answer. For example, recent methods prompt LLMs to produce Python programs for arithmetic or logical reasoning problems, thereby delegating computation to a reliable interpreter and achieving significantly higher accuracy than standard chain-of-thought reasoning alone (Chen et al. (2022); Gao et al. (2023)). Huang et al. (2025) use LLMs to generate program-based labelers for weak supervision. In this work, we employ the coding ability of LLMs to generate implementations for SPTs.

## 3 PRELIMINARIES

### 3.1 NOTATION

Let $\mathcal{X}$ be the space of programs and $\mathcal{T} \subseteq \mathcal{X}^{\mathcal{X}}$ the space of program transformations. A functional equivalence oracle $A : \mathcal{X} \times \mathcal{X} \to \{0, 1\}$ is a binary function that evaluates if two programs are semantically equivalent. A common implementation of an equivalence function is to verify if the two programs produce identical outputs for a set of test inputs. Let $\mathsf{T} \in \mathcal{T}$ be a *semantics-preserving transformation* that perturbs a program but preserves the semantics, i.e. $\mathsf{T} : \mathcal{X} \to \mathcal{X}$ $s.t.$ $\forall x \in \mathcal{X}, A(x, \mathsf{T}(x)) = 1$. Single SPTs can be combined to form a set $T \subset \mathcal{T}$. Now, we define $M : \mathcal{X} \times \mathcal{X} \to [0, 1]$ as a *clone-detector model* that outputs the likelihood of two input programs being functional equivalents of each other. The binary classification (clone vs. non clone) is usually done using a threshold. A perfect clone detector would be an equivalence oracle, but that is unrealizable due to the undecidability of program equivalence (Rice (1953)).

### 3.2 WORST-CASE TRANSFORMATIONS FOR CLONE-DETECTION MODELS

For clone detectors, a worst-case transformation is a functionally equivalent program that is classified as a non clone. We represent this problem as finding a semantic adversarial perturbation against a clone-detector model. For a program $x \in \mathcal{X}$ and a clone-detector model $M$, we can find a transformed $x'$ by solving the following optimization:

$$\underset{x' \in \mathcal{X}}{\arg\max} \, L(x, x') \quad s.t. \quad A(x, x') = 1, \tag{1}$$

where $L(x, x') = 1 - M(x, x')$ is the distance between programs $x$ and $x'$ according to the clone detector. A solution to this optimization problem needs to be a program that is (1) valid (syntactically correct) and (2) functionally equivalent to the original program $x$. Solving this problem, while satisfying these requirements, is challenging because of the complexity of both the search space and the equivalence constraint.

### 3.3 SEMANTICS-PRESERVING TRANSFORMATIONS (SPTs) AND TRANSFORMATION SETS

Semantics-preserving transformations provide a way to enumerate programs satisfying the *validity* and *equivalence* requirements. These transformations operate through symbolic manipulations of an input program and are carefully designed to preserve the program's semantics. For example, consistently renaming all occurrences of a variable throughout its scope is a basic SPT, satisfying both validity and equivalence. Given the set of SPTs: $T \subset \mathcal{T}$, we can rewrite Equation 1 as finding a sequence of transforms:

$$(\mathsf{T}_k, \ldots, \mathsf{T}_1) = \arg \max_{\mathsf{T}_k, \ldots, \mathsf{T}_1 \in T} L(x, (\mathsf{T}_1 \circ \cdots \circ \mathsf{T}_k)(x)), \tag{2}$$

where the transformed code is $x' = (\mathsf{T}_k \circ \cdots \circ \mathsf{T}_1)(x)$, and $k$ is the length of the transformation sequence.

### 3.4 STRENGTH AND DIVERSITY OF TRANSFORMATION SETS

Finding a worst-case transformation that is successful against a clone detector (i.e., a solution to Equation 1) corresponds to finding a set of *diverse* SPTs of sufficient *strength*. Strength, in this context, refers to the degree to which a transformation increases the distance between the original program and transformed program according to the clone detector. Transformations that induce significant alterations to the code's structure, complexity, or predictability tend to be stronger. For instance, stronger transformations often modify larger sections of the code or fundamentally change its control or data flow patterns. SPTs like converting **for** loops to **while** or related control-flow constructs, and function in-lining are stronger whereas swapping independent instructions, variable renaming, or changing the order of operands in an expression are examples of weaker SPTs. Additionally, diversity of SPTs is important: relying solely on a single type of transformation, even a strong one, is insufficient to effectively explore the vast space of equivalent programs and solve Equation 1. This limitation stems from the specialized nature of individual SPTs, which typically target only specific aspects of the program.

We capture this intuition by providing an upper bound on the strength of a sequence of transformations as a function of their diversity. First, we define the diameter of a transformation.

**Definition 3.1.** (Diameter of a transformation set.) The $k$-step diameter of a set of transformations $T \subset \mathcal{T}$ is

$$\mathcal{D}_k^T(x) = \max_{\substack{\mathsf{T}_1, \ldots \mathsf{T}_k \in T \\ \mathsf{T}_1', \ldots \mathsf{T}_k' \in T}} L((\mathsf{T}_k \circ \ldots \circ \mathsf{T}_1)(x), (\mathsf{T}_k' \circ \ldots \circ \mathsf{T}_1')(x)).$$

Second, we introduce a measure to quantify the diversity of SPT sets. This measure captures the increase in the diameter of a transformation set when adding a new transformation.

**Definition 3.2.** (Diversity of an SPT set.) The $k$-step diversity of a set of semantics-preserving transformations $T \subset \mathcal{T}$ is given by

$$\Delta_k^T(x) = \frac{\mathcal{D}_k^T(x)}{\mathbb{E}_{t \in T} \mathcal{D}_k^{T \setminus t}(x)}.$$

Third, we observe that the strength of transformation achievable through the application of $k$ SPTs from a given set is upper bounded by the diversity of its subsets.

**Lemma 3.3.** *Upper bound for $k$-step transformation strength.*

$$\max_{\mathsf{T}_1, \ldots, \mathsf{T}_k \in T} L(x, (\mathsf{T}_1 \circ \cdots \circ \mathsf{T}_k)(x)) \leq \prod_{i=1}^{k} \Delta_k^{T_i}, \quad where \ T_i := T_{i-1} \cup \{\arg \max_{t \in T \setminus T_{i-1}} \mathcal{D}_k^{T_{i-1} \cup t}(x)\}$$

**Limitations of Existing SPTs**  SPTs have been explored for many applications, including robustness eval-uation and data augmentation for code models. Most approaches only consider simple SPTs, like swapping independent code blocks, replacing while loops with for loops, switching if conditions, and renaming vari-ables, in isolation. While some do perform compositions of individual SPTs, they only consider a small number of distinct transformation types. For instance, the prior works summarized in Table 5, in total, uti-lize a limited set of *eight* distinct SPTs. To automatically perform these transformations, existing approaches use heuristic based or ad-hoc implementations that may not apply the strongest version of that transformation type. As a result of Lemma 3.3, existing SPTs, even if composed, are neither strong nor diverse enough to yield optimal solutions for the problem in Equation 1. We confirm this observation empirically in section 5, where we show SPTs from existing approaches are not strong against code clone detectors.

## 4 THE Auto-SPT FRAMEWORK

We propose Auto-SPT, an LLM-based automation framework to (1) design diverse SPTs, (2) generate their strong implementations, and (3) combine them to create stronger transformations.

### 4.1 STEP 1: DESIGNING NEW SPTS

We use LLMs to automatically design new SPTs. However, as highlighted by previous work (Chen et al. (2024); Hooda et al. (2024b)), naively prompting the LLMs does not work. LLMs suffer from a lack of randomness (Zhang et al., 2024), which restricts the diversity of the designed transformations. Moreover, LLMs are prone to hallucinations, which can affect the correctness of transformations. We address these problems through iterative prompting, where we employ a specially crafted *generator prompt*. This prompt instructs the LLM to generate SPTs that are distinct from the list of transformations generated so far. This instruction helps to guide the generation of new and diverse transformation. We use this prompt template:

> You are a python programming language expert. Your goal is to design new semantic preserving transfor-mations. I will give you 5 python programs and you have to suggest a transformation that can be applied to all the 5 programs. Give an exact description of the transformation such that it can be used to implement the transformation. The output format should be:
>
> - Transformation Name: `<name>`
> - Description: `<description>`
>
> The transformation should be distinct from the list of following transformations:
> {transformation_list}
> PROGRAMS:
> {programs}

Table 1 lists a set of 20 new SPTs generated in this step. As evident from the table, these transformations target different aspects of the code, including arithmetic operations, data structures, control flow, function execution, and variable names. They target both low-level computation (e.g., arithmetic and bitwise manipu-lations) and higher-level program behaviors (e.g., nested functions, deferred execution, generator chaining).

### 4.2 STEP 2: IMPLEMENTING SPTS

After designing the SPTs, the next step is to apply them to programs. We prompt an LLM to automatically generate a (source code) implementation of a desired SPT. We directly use the transformation descriptions generated in the previous stage and use temperature sampling to generate multiple candidate implementa-tions for each SPT. We then use a subset of 1000 randomly selected programs (with accompanying unit

Table 1: New Semantic Preserving Transformations. The number in the braces shows correctness+applicability on 1000 programs. Although some of the transformations are only applicable to a small set of programs, each program has atleast 9 applicable transformations. We check for correctness by running unit tests, and applicability by ensuring that the transformed program is not identical to the original.

| | |
|---|---|
| Modular Arithmetic Distribution (568) | Temporal State Inversion with Deferred Execution Queue (229) |
| Nested Function Encapsulation with Dynamic Execution (615) | Dimensional Array Flattening with Dynamic Offset Mapping (272) |
| Recursive List Comprehension Flattening (128) | Probabilistic Control Flow Injection with State Preservation (264) |
| Modular Index Obfuscation with Dynamic Range Mapping (471) | Circular Buffer State Threading with Modular Iteration (412) |
| Cyclic Variable Rotation with Lambda Chaining (11) | Iterative State Composition with Delayed Resolution (91) |
| Conditional Expression Vectorization with Boolean Masking (193) | Arithmetic Expression Decomposition with Modular Chain Substitution (439) |
| State Variable Accumulation with Bitwise Memory (840) | Modular State Interleaving with Conditional Aggregation (97) |
| Exponential Base Conversion with Dynamic Radix Chaining (398) | Modular State Accumulation with Circular Buffer Mapping (713) |
| Modular State Threading with Generator Chaining (649) | Cumulative State Mapping with Modular Dictionary Encoding (141) |
| State Machine Enumeration with Dynamic Dispatch (13) | Positional State Encoding with Circular Dependency (85) |

tests) to estimate the correctness and strength of the candidate implementations. For this set of validation programs $\{x_1, ..., x_n\}$ and a transformation candidate $T \in \mathcal{T}$, we use the following reward function:

$$R(T) = \frac{1}{n} \sum_{i=1}^{n} A(x_i, T(x_i)) \cdot L(x_i, T(x_i)),$$

where $A : \mathcal{X} \times \mathcal{X} \to \{0, 1\}$ checks for functional equivalence by executing unit tests. We use this reward function for our Best-of-N sampling to select the strongest implementation for each SPT. We use the following prompt template.

> Write a python program that takes in a string (from std input) that represents another python program, mutates it according to the following transformation and prints the result (do not print anything else).
> Transformation: {transformation description}
>
> - the obfuscator should ensure that the program is still valid python code
> - the obfuscator should be semantic preserving
> - remember to input the entire program which can include multiple lines

Table 3 compares the applicability and correctness of the automatically generated transformations from this step to manually implemented ones. We check for correctness by running unit tests, and applicability by ensuring that the transformed program is not identical to the original. Measured over a random sample of 1000 programs (that were not part of the generation prompt), the automatically generated transformations by Auto-SPT apply and result in correct behavior to roughly the same number programs as those manipulated through manual transformation. This result highlights the effectiveness of this approach in automatically transforming code. Further, Table 1 shows the applicability and correctness of the new SPTs. This table highlights that the new SPTs apply to varied sets of programs, according to the coding style and implementations. Although some of the transformations are only applicable to a small set of programs, each program has atleast 9 applicable transformations.

In conclusion, this approach has three benefits. First, it automatically generates the transformation code, avoiding the overhead of manual implementation from prior work. Second, it finds a strong implementation of a given transformation. Third, it obviates the need to prompt the LLM to apply the transformation for each code sample. Doing so would incur prohibitive cost and affect the consistency of applied transformations.

### 4.3 STEP 3: COMBINING SPTS

Solving the optimization problem in Equation 2 requires searching the potentially vast space of programs $x'$ that are semantically equivalent to the original program $x$. We employ an iterative beam search strategy,

guided by the clone detector score $M$ and utilizing a set of available SPTs $\mathcal{T} = \{T_1, T_2, \ldots, T_N\}$. This search maintains a beam of the $B$ most promising candidate transformation at each step, starting with just the original program $x$. Each iteration involves three stages. The *expansion* stage applies every available SPT from the set $\mathcal{T}$ to each of the current best programs residing within the beam to expand the set of next-step candidates. The *filtering* stage prunes the expanded pool to ensure they syntactically valid and semantically equivalent to the original program. The third stage, *selection*, chooses top $B$ candidates that are most distant from the original program using the clone detector. This cycle of expansion, filtering, and selection repeats for a predefined number of iterations $N$. The process terminates by selecting the single program from the final beam that achieved the overall lowest score $M(x, x')$ during the search, representing the best transformation found. Algorithm 1 describes the procedure in detail.

## 5 EXPERIMENTS

Our evaluation answers three key research question:

**RQ1:** How effective is `Auto-SPT` at evading clone detectors as compared to existing transformations?

**RQ2:** Why are `Auto-SPT`'s SPTs more effective than existing transformations?

**RQ3:** Do `Auto-SPT`'s SPTs help train more robust clone detectors?

### 5.1 SETUP

**Dataset.** We use the CodeContests dataset from Li et al. (2022) to construct clone and non-clone code pairs. Each problem in this dataset comes with multiple solutions (submitted by actual users via platforms like Codeforces, CodeChef, etc.) as well as unit tests. We use solutions belonging to the same problem to construct clone pairs and those belonging to distinct problems to construct non-clone pairs. In total, we randomly select 2000 distinct problems. Then, we split this to get 1500 training, 250 validation, and 250 test sets of problems. The split is done to test the generalization performance of the clone detection task across different functionalities. For each set, we construct 250 clone pairs per problem, leading to a total of 375000 clone pairs in the training set. We construct an equal number of non-clone pairs to get a balanced dataset. We ensure that we only consider solutions that pass all the unit tests associated with the corresponding problem. Our evaluation focuses on Python programs, but our method should apply to any programming language.

**Clone Detection Models.** We evaluate `Auto-SPT` using four different model architectures – Code-BERT (Feng et al., 2020b), GraphCodeBERT (Guo et al., 2021), EmbeddingGemma (Gemma-Team et al., 2025), and Snowflake's Arctic Embed M (Merrick et al., 2024). We perform full finetuning for the Code-BERT model using the training set and select the best checkpoint based on the accuracy of the validation set. For the other models, we use the pretrained weights and fine-tune the final layer.

**`Auto-SPT` Hyperparameters.** We use Gemini 2.5 Pro to design and implement SPTs (i.e., as the LLM for Steps 1 and 2 of `Auto-SPT`). We use a temperature of $0.1$ for the design step and $0.8$ for the implementation step. We use the remaining solutions from CodeContest (disjoint from the 2000 problems considered during dataset generation) to populate the example programs in the generator prompt. We use $N = 20$ for Best-of-N sampling during the implementation step. Again, we sample the validation programs to estimate the reward function from the remaining CodeContest solutions. For the beam search, we use a beam size of 5 and apply each SPT to all the candidates in the beam. We use the following notation to denote different transformation sets. `VarRename`, `Conditional`, `ForWhile` and `IfElseFlip` are together described as `Orig-4`. We use `Auto-SPT` to generate 5 different implementations for each of the above four transformations, which is described by `Orig-20`. The transformations mentioned in Table 1 are represented by `New-20`.

Table 2: Worst Case Transformation for different transformation sets when searching over 10 iterations. `Auto-SPT` achieves the strongest transformations.

| Clone Detection Model | Worst Case Distance | | | |
|---|---|---|---|---|
| | Wang et al. (2022) | Auto-SPT (Orig-4) | Auto-SPT (Orig-20) | Auto-SPT (New-20) |
| CodeBERT | 0.243 ± 0.395 | 0.514 ± 0.467 | 0.533 ± 0.484 | **0.947 ± 0.218** |
| GraphCodeBERT | 0.092 ± 0.229 | 0.115 ± 0.266 | 0.147 ± 0.279 | **0.721 ± 0.319** |
| EmbeddingGemma | 0.285 ± 0.369 | 0.292 ± 0.378 | 0.390 ± 0.434 | **0.803 ± 0.383** |
| snowflake-arctic-embed-m-v2.0 | 0.067 ± 0.193 | 0.097 ± 0.215 | 0.194 ± 0.325 | **0.929 ± 0.227** |

**Compute.** We run all experiments on a machine with 4 NVIDIA H100 GPUs, 40 Intel(R) Xeon(R) Silver 4410T CPUs, and 126GB of RAM. More details on compute requirements of `Auto-SPT` are in Appendix C.

## 5.2 RQ1: ATTACK EFFECTIVENESS

Table 2 shows distance measured by the clone detector model for the worst-case transformation against four different types of clone detection models. For the search, we consider four different settings which correspond to four different sets of SPTs and their implementations –Wang et al. (2022) (`Orig-4`), `Auto-SPT` (`Orig-4`), `Auto-SPT` (`Orig-20`) and `Auto-SPT` (`New-20`), where the third type is constructed by taking distinct implementations (5 each) of the `Orig-4` transformations. Across the three models, we make two main observations: (1) For the same type of transformations, `Auto-SPT` outperforms implementations provided by past work, and (2) SPTs designed by `Auto-SPT` can be composed to perform stronger transformations as compared to existing SPTs. Interestingly, even recent and state-of-art code embedding models are not robust against transformations from prior work and SPTs from `Auto-SPT`.

## 5.3 RQ2: TRANSFORMATION SET DIVERSITY

Table 2 reveals an important observation: improving diversity of transformations makes them stronger. We can observe this along two dimensions – (1) Increasing the diversity of implementations for the same set of SPTs improves transformation strength (`Auto-SPT Orig-20` > `Auto-SPT Orig-4`), (2) Increasing the diversity of the types of SPTs further makes transformations stronger (`Auto-SPT New-20` > `Auto-SPT Orig-20`). To delve deeper into this observation, we use the search procedure described in subsection 4.3 to estimate the diversity (Definition 3.2) of the sets of transformations. In Table 4, we show the diversity values for an increasing number of implementations from two different sets of transformations – `Orig-4`, and `New-20`. To expand the `Orig-4` set to 20 implementations, we use `Auto-SPT` to generate multiple implementations for each SPT type. Due to the high computation cost of estimating diversity, we limit the maximum number of search iterations to 5 (since estimating diversity for a set of $k$ transformations requires running $k$ search procedures). We see that the diversity of `Orig-4` goes down rapidly as the number of transformations increases. Going from 12 to 16 transformations has a minimal effect on the transformation strength. On the other hand, `New-20` maintains diversity even for 20 transformations.

## 5.4 RQ3: EVALUATING ROBUST DETECTORS

Next, we evaluate whether SPTs from `Auto-SPT` can improve the robustness of clone detectors. We choose to perform this experiment on CodeBERT because it provides official support for finetuning for the clone detection task (Feng et al., 2020a). We finetune CodeBERT using training data generated by applying transformations from `Orig-4`. We consider two settings – SPT implemented by Wang et al. (2022) and `Auto-SPT`. We transform each training point with a probability of 0.5, and then select a random SPT from the transformation set. Figure 1 shows the transformation effectiveness for all three settings. First, we find that training

Table 3: Existing Semantic Preserving Transformations.

| Transformation | Method | Correct + Applicable (1000 programs) |
|---|---|---|
| VarRename | Wang et al. (2022) | 989 |
| | Auto-SPT | 760 |
| Conditional | Wang et al. (2022) | 680 |
| | Auto-SPT | 458 |
| ForWhile | Wang et al. (2022) | 621 |
| | Auto-SPT | 706 |
| IfElseFlip | Wang et al. (2022) | 241 |
| | Auto-SPT | 218 |

Table 4: Diversity for increasing number of transformations. The target clone detector model is CodeBERT.

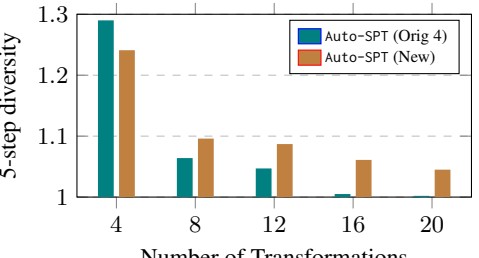

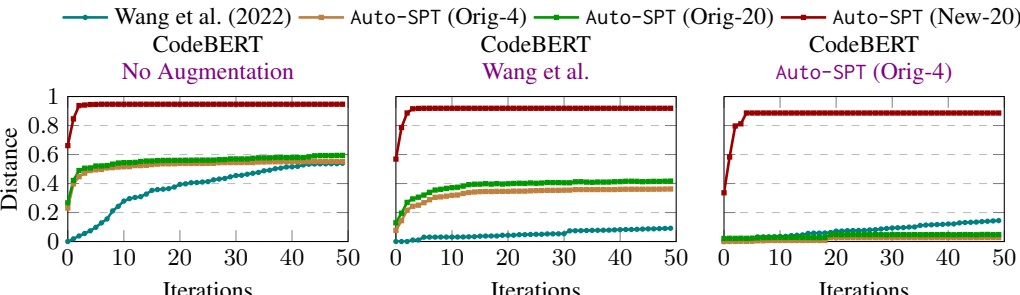

Figure 1: Effectiveness of different transformation sets against clone detectors trained to be robust to SPTs. Auto-SPT provides stronger transformations which lead to more robust clone detection models. The text in violet describes the setting used to generate transformations for training.

on transformations from Wang et al. (2022) improves robustness the most against the same transformation set. It provides minor improvements against Orig-4 or New-20 transformations from Auto-SPT. However, training on Orig-4 transformations from Auto-SPT improves robustness against both Wang et al. (2022) and Auto-SPT's implementations. It slightly improves against New-20 transformations, suggesting that training on stronger transformations helps improve robustness against weaker transformations of the same type.

## 6 CONCLUSION

In this paper, we presented Auto-SPT, a novel automated framework leveraging large language models to generate diverse and strong semantic-preserving transformations (SPTs). Auto-SPT addresses the limitations of existing heuristic-based and manually implemented SPTs by automating the design and implementation of transformations, thus significantly reducing manual effort and improving transformation strength and diversity. Our theoretical analysis formalizes the relationship between transformation diversity and obfuscation strength, while our empirical evaluation demonstrates that Auto-SPT-generated transformations significantly degrade the performance of state-of-the-art code clone detectors compared to existing approaches.

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

# A  EXISTING SEMANTIC PRESERVING TRANSFORMATIONS

Table 5: Existing Semantic Preserving Transformations.

| Past Work | Transformations | | | | | |
| --- | --- | --- | --- | --- | --- | --- |
| | VarRename | Conditional | ForWhile | IfElseFlip | StmtSwap | Unroll |
| Allamanis et al. (2021) | ✓ | ✓ | | ✓ | | |
| Rabin et al. (2021) | ✓ | ✓ | ✓ | | ✓ | |
| Bui et al. (2021) | ✓ | | ✓ | | ✓ | |
| Henkel et al. (2022) | ✓ | ✓ | | | | ✓ |
| Chakraborty et al. (2022) | ✓ | | ✓ | ✓ | | |
| Yang et al. (2022) | ✓ | | | | | |
| Zhang et al. (2022) | ✓ | | | | | |
| Wang et al. (2022) | ✓ | ✓ | | ✓ | | |
| Hooda et al. (2024a) | ✓ | | | ✓ | ✓ | |
| Le-Cong et al. (2024) | | ✓ | | ✓ | | |
| Dong et al. (2024) | ✓ | ✓ | | | | |

# B  COMPOSING TRANSFORMATIONS

---

**Algorithm 1** Searching for the Strongest Composition of Transformations

---

**Input:** Original program $x \in \mathcal{X}$, Set of SPTs $\mathcal{T} = \{T_1, \ldots, T_k\}$, Equivalence checker $A : \mathcal{X} \times \mathcal{X} \to \{0, 1\}$, Clone detector $M : \mathcal{X} \times \mathcal{X} \to [0, 1]$, Beam size $B \in \mathbb{N}^+$, Number of iterations $N \in \mathbb{N}$
**Output:** Worst Case Transformation $x'_{best} \in \mathcal{X}$ such that $A(x, x'_{best}) = 1$
1:  Beam$_0 \leftarrow \{x\}$
2:  **for** $i = 0$ to $N - 1$ **do**
3:      Candidates$_i \leftarrow \emptyset$
4:      **for** $x_j \in$ Beam$_i$ **do**
5:          **for** $T_k \in \mathcal{T}$ **do**
6:              Candidates$_i \leftarrow$ Candidates$_i \cup \{T_k(x_j)\}$                    ▷ Expand beam
7:          **end for**
8:      **end for**
9:      FilteredCandidates$_i \leftarrow \emptyset$
10:     **for** $x' \in$ Candidates$_i$ **do**
11:         **if** IsValid($x'$) **and** $A(x, x') = 1$ **then**                    ▷ Filter for validity and equivalence
12:             FilteredCandidates$_i \leftarrow$ FilteredCandidates$_i \cup \{x'\}$
13:         **end if**
14:     **end for**
15:         ▷ Select B candidates that best minimize M
16:     Beam$_{i+1} \leftarrow$ top$_B$(FilteredCandidates, min $M(x_i, x)$)
17: **end for**
18: $x'_{best} \leftarrow \underset{x' \in \text{Beam}_N}{\operatorname{argmin}} M(x, x')$                    ▷ Select final best
19: **return** $x'_{best}$

---

# C  COMPUTE REQUIREMENTS OF Auto-SPT

Analyzing the cost of Auto-SPT involves looking at two separate stages:

Generating transformation implementations: This stages only needs to happen once and then the generated transformation implementation can be deterministically used to transform a large number of programs without requiring any LLM calls. As described in Section 4.1, we make 20 LLM calls to design 20 SPTs using

the generator prompt. Then, we generate 20 implementations each (for best of N selection) for these 20 SPTs, resulting in a total of 400 LLM calls. Assuming an average input and output context of 10k tokens each for these LLM calls (all our prompt context and generated outputs are well below 10k tokens each), we get a total estimate of less than $50 for Gemini-2.5 Pro. Each of these 400 LLM calls can be done in parallel, and each only takes around 1 minute to complete.

Applying and searching for transformation compositions: Applying the transform, checking for functional equivalence and computing the reward for one candidate takes around 0.04 seconds for GraphCodeBERT. This means that for beam size 5 and transformation set of size 4, each iteration takes around 0.8 seconds. And therefore, the search for 50 steps takes around 40 seconds.

