# OpenReview forum: "Auto-SPT: Automating Semantic Preserving Transformations for Code"
_ICLR.cc/2026/Conference — Submitted to ICLR 2026_

### Official Review · Reviewer_MX1P · 2025-10-30

**Soundness:** 3
**Presentation:** 2
**Contribution:** 3
**Rating:** 6
**Confidence:** 4

**Summary:**

This paper aims to address a gap in the field of code clone detection, where models are typically trained on clean and structured data but struggle in real-world scenarios where code undergoes a variety of Semantic Preserving Transformations (SPTs).
To address this gap, the authors introduce Auto-SPT, a novel framework that automatically generates stronger SPTs, which can be used to train more robust clone detectors or to evaluate the robustness of clone detector models.
The authors formally define the concept of diversity for SPTs and show that diversity is a key factor in producing strong transformations. They further demonstrate that state-of-the-art clone detectors perform poorly on transformations generated by Auto-SPT, revealing a lack of robustness for challenging clones.

**Strengths:**

1. This work provides a rigorous formalization for constructing worst-case transformations for clone detectors — i.e., programs that are semantically equivalent but not recognized as clones.
This formalization helps identify two key properties of strong SPTs: strength (measured as the distance between original and transformed programs) and diversity. Results in Table 2 and Figure 1 clearly demonstrate the importance of these properties.
2. The proposed framework, powered by LLMs, automates the design, implementation, and composition of SPTs. This is a novel contribution, as previous methods relied on heuristics and required time-consuming, ad hoc implementations.
Auto-SPT generates 20 new SPTs, as well as alternative implementations of existing ones, showing that both approaches lead to more challenging transformations.
3. The paper introduces a new method for combining SPTs using beam search instead of heuristic methods, improving both strength and diversity.
4. Table 2 convincingly shows that even recent clone detectors struggle with clones generated by Auto-SPT. In particular, the New-20 set could serve as a strong benchmark to test the robustness of Code LLMs in distinguishing semantically equivalent code.

**Weaknesses:**

1. Lack of qualitative analysis: examples of programs generated by Auto-SPT are not shown in the main paper nor in the appendix. Similarly, a more quantitative analysis on what makes these SPTs stronger would help to understand more the differences with previous approaches.
2. Although a reasonable number of clone detection models are evaluated, several modern frontier models (e.g., GPT-5, Claude Sonnet 4.5), or open-source models (e.g., Qwen, DeepSeek) are missing. Would these models struggle as well with programs from Auto-SPT?
3. The framework could indeed be used to measure model robustness against adversarial clones. However, it is less clear how Auto-SPT can be used for training. Training experiments are limited to CodeBERT in Python: generalizability of the framework could be improved by training different models and expanding to more languages (there are code clone benchmarks for C++ and Java, as well as data sources similar to CodeContest). Also, it is not shown whether training on Auto-SPT programs might harm other coding capabilities of the LLM.
4. While Figure 1 and Table 2 demonstrate that Auto-SPT-generated clones are challenging across models, the paper lacks a final experiment showing that a model trained on Auto-SPT-transformed programs also performs well on standard clone detection benchmarks.

**Questions:**

Questions related to weakness #1:
- Why are programs generated from Auto-SPT more challenging for clone detectors?
- How do these generated clones differ from those produced in previous papers?
- Which part of the pipeline - design, implementation, or combination - makes Auto-SPT programs so challenging?
- Do Auto-SPT programs look more natural (i.e., as code clones we could find in human written repositories)?

Questions related to weakness #3:
- Why is training limited to Auto-SPT (Orig-4) in Figure 1? If CodeBERT were trained on the New-20 subset, would it perform better on a test set drawn from the same distribution?

Minor comments:
- line 250: missing closing “)”
- Sometimes the paper is not straightforward to read: I think the definition of Orig-4, Orig-20 and New-20 should be made more clear. Similarly, the last part of the introduction (line 070) can be understood only after reading the experimental part of the manuscript.

---

### Official Review · Reviewer_UxyX · 2025-10-31

**Soundness:** 3
**Presentation:** 3
**Contribution:** 2
**Rating:** 2
**Confidence:** 4

**Summary:**

This paper proposes Auto-SPT, a framework for automatically designing, implementing, and writing various semantically preserving transformations (SPTs) to evaluate and enhance the robustness of code clone detection models. The authors formalize the concepts of transformation strength and diversity and empirically demonstrate that their automated framework can generate more effective transformations than existing heuristic-based methods. Using these generated transformations for data augmentation can improve model robustness.

**Strengths:**

- The paper provides a formal analysis linking SPT diversity to transformation strength, with empirical validation on real datasets.

**Weaknesses:**

The novelty is limited. Previously work, such as (Zhang et al., 2023), already described semantic-preserving code transformations for
evaluating machine learning-based code clone detection models. That paper also presented many semantic preserving transformations as well as their combinations. This paper only briefly mentioned (Zhang et al., 2023) but did not compare with it.
Also, there are some recent work for improving the robustness of the clone detection model, which can be compared with too. For example:

[1] Tian Z, Chen J, Jin Z. Code difference guided adversarial example generation for deep code models, ASE 2023.

[2] Huang L, Sun W, Yan M. Iterative Generation of Adversarial Example for Deep Code Models, ICSE 2025.

Furthermore, in lines 159-162, the author claims that SPTs, such as converting a for loop to a while loop, are stronger, while variable renaming is a weaker SPT. This claim lacks empirical evidence and additional experiments are needed to support it. In fact, research has shown that variable renaming can induce model errors effectively (see the above two references).

RQ2 and RQ3 involve too few models (only CodeBERT), which is not enough to support the conclusion. Also, many code models (such as CodeBERT and GraphCodeBERT) are quite old. Many recent, SOTA code models are not evaluated.

In terms of presentation, it would be better to present real code samples that the new SPT generates, particularly for the 20 new SPTs listed in Table 1.

The paper uses unit tests to verify the preserved semantics of the generated code, it is not clear after drastic transformations (especially the combinatorial transformations), if unit tests can still perform complete semantic verification.

**Questions:**

How is the proposed approach compared with the related work mentioned in the Weaknesses?

---

### Official Review · Reviewer_zt3M · 2025-11-01

**Soundness:** 2
**Presentation:** 3
**Contribution:** 2
**Rating:** 2
**Confidence:** 3

**Summary:**

This paper investigates the performance degradation of clone detection models after deployment when facing test data distribution shifts. The authors leverage LLMs to generate more diverse code transformations and use these transformations to create test cases that are not present in the training data. The paper validates how clone detection models behave when encountering out-of-distribution cases. Experimental results demonstrate that the proposed approach can generate clone code variants that break existing clone detection models.

**Strengths:**

1. Clone detection is an important problem in software engineering, and exploring distribution shift after model deployment is valuable.
2. The paper achieves automatic discovery of new code transformations using LLMs.
3. The results show that LLM-based data augmentation can generate realistic cases that exist in practice but are missing in training data.

**Weaknesses:**

Method:

1. Lines 207–209 mention that naively prompting the LLMs does not work due to lack of randomness and hallucinations. The authors propose a new prompt design, but it is not clearly explained why the proposed prompt can address these two issues, nor is there empirical evidence demonstrating that the prompt resolves them.
2. Verifying whether the transformations can be correctly applied is critical (one of the limitations of prior work mentioned in the introduction). In Section 4.2, the authors use “correctness + applicability” as a joint metric to validate transformations. However, correctness and applicability should be evaluated separately to better understand whether a transformation can produce valid code. Applicability only reflects generality, while correctness reflects whether the output code is valid. For example, if only 9 out of 1000 transformed examples pass correctness + applicability, but most failures are due to applicability, the transformation may still be useful; if failures are due to incorrect code generation, the transformation is of poor quality. The authors should analyze correctness and applicability separately.

Evaluation:

1. Dataset. The authors should more clearly justify the choice of CodeContests, e.g., whether it is widely used in prior clone detection research. Validation on a single dataset is limited. Common datasets in prior work include OJClone [1], GCJ [2], and BCB [3]. The paper should report results on these datasets or clearly explain why they are not applicable.
2. Baseline. Only one baseline is used, which is insufficient to convincingly demonstrate the effectiveness of the proposed approach. The appendix lists eight related works; more baselines should be included, or the paper should justify why they cannot be compared.
3. Validity. Although the method verifies the correctness of generated transformations on 1000 code samples, the correctness of transformed code in the evaluation stage is also critical. The paper should include validation of the correctness of the transformed code used in experiments.
4. Hyperparameters. The choice of hyperparameters (e.g., temperature) needs clearer justification, including their origin and impact on results.


Line 348: “across the three models” — should this be four?

References
[1] L. Mou, G. Li, L. Zhang, T. Wang, and Z. Jin, “Convolutional neural networks over tree structures for programming language processing,” inProceedings of the AAAI conference on artificial intelligence, vol. 30,
no. 1, 2016.
[2] “Google Code Jam,” https://code.google.com/codejam/contests.html, 2016, accessed: 2016-10-8.
[3] J. Svajlenko, J. F. Islam, I. Keivanloo, C. K. Roy, and M. M. Mia, “Towards a big data curated benchmark of inter-project code clones,” in 2014 IEEE International Conference on Software Maintenance and Evolution. IEEE, 2014, pp. 476–480.

**Questions:**

1. How does the designed prompt address the LLM issues of insufficient randomness and hallucinations in transformation generation?
2. Does combining correctness and applicability into one metric affect the ability to evaluate transformation quality?
3. Are additional datasets and baselines not applicable to this work? If so, why?
4. Were all the generated transformed code samples correct during experiments?
5. Why were the chosen hyperparameters used, and how do they influence the results?

**Details Of Ethics Concerns:**

N.A.

---

### Official Review · Reviewer_95aw · 2025-11-02

**Soundness:** 2
**Presentation:** 2
**Contribution:** 2
**Rating:** 2
**Confidence:** 5

**Summary:**

This paper studies semantic-preserving transformations (SPTs) and proposes Auto-SPT, a framework for automatically constructing synthetic data generators for code. The authors use LLMs to design new SPTs and to generate corresponding implementation scripts that automatically perturb Python functions. They then finetune several embedding models on data generated by Auto-SPT and compare performance against baseline methods. Experimental results show that Auto-SPT can effectively degrade clone detector performance, demonstrating its potential as a robustness evaluation tool.

**Strengths:**

- The correctness of the generated transformations is validated through corresponding unit tests, which enhances the reliability of the proposed SPTs.

- The framework is largely automated and demonstrates good scalability for producing diverse code transformations.

**Weaknesses:**

- The automation in Auto-SPT primarily relies on prompting LLMs to design and implement SPTs. While practical, this approach offers limited methodological novelty given the growing application of LLM-based automation.

- The framework currently supports only Python and focuses on function-level transformations, while file- or project-level perturbations would better reflect real-world code evolution.


- The paper does not evaluate whether the transformed programs remain natural and consistent with human coding practices, which is important for assessing the realism of generated data.

**Questions:**

- How are the LLM prompt templates designed? Were there iterative refinements or prompt-tuning steps to improve transformation quality?

- The reward function relies on clone detector model scores. How is the reliability of this clone detector ensured or validated?

---

### Meta-Review · Area_Chair_BiNx · 2026-01-01

**Summary:**

This paper proposes Auto-SPT, an automated framework for generating semantic-preserving code transformations using large language models, with the goal of evaluating and improving the robustness of code clone detection models under real-world distribution shifts. Reviewers generally agree that the problem is important and practically motivated, and that semantic-preserving transformations are a relevant tool for stress-testing clone detectors beyond clean training data. Several reviewers also acknowledge that the framework is largely automated, scalable, and supported by empirical results showing significant performance degradation of existing clone detection models under Auto-SPT-generated transformations.

However, reviewers raised substantial concerns regarding limited methodological novelty, insufficient evaluation breadth, and unclear experimental validation of key claims. Notably, the authors did not submit a rebuttal, and therefore none of these concerns were addressed or clarified during the response phase. As a result, the assessment necessarily reflects only the originally submitted manuscript and the reviewers’ initial evaluations, which informed the final recommendation.

**Reviewer Concerns:**

Across reviews, there is broad agreement that Auto-SPT addresses a meaningful gap between training and deployment conditions for code clone detection models, and that automating the generation and composition of semantic-preserving transformations using LLMs is practically appealing. Reviewers highlighted strengths such as the use of unit tests to validate transformation correctness, the formal discussion of transformation diversity and strength, and empirical evidence that Auto-SPT-generated code can expose robustness weaknesses in existing clone detectors.


At the same time, reviewers consistently raised multiple core concerns that remain fully outstanding due to the absence of an author rebuttal. Several reviewers questioned the novelty of the approach, noting that prior work has already explored semantic-preserving code transformations and their combinations, and that the primary contribution here lies in automating this process with LLMs rather than introducing fundamentally new transformation principles. Reviewers also emphasized that the evaluation is limited in scope: experiments focus primarily on Python, function-level transformations, a small number of clone detection models (often centered on CodeBERT), and a single main dataset, with insufficient justification for excluding widely used benchmarks and stronger baselines from prior literature.


In addition, concerns were raised about experimental validity and analysis, including the conflation of correctness and applicability into a single metric, the lack of separate analysis of transformation failure modes, limited qualitative examples of generated code, and unclear justification of prompt design and hyperparameter choices. Some reviewers further questioned whether unit tests alone are sufficient to guarantee semantic preservation under aggressive or compositional transformations, and whether the generated code remains natural and representative of real-world code evolution. Since no rebuttal was provided, these issues could not be examined further or mitigated.

**Reviewer Scores:**

Given that no rebuttal was submitted, reviewers did not have an opportunity to reconsider their assessments in light of clarifications, additional experiments, or scope adjustments. Consequently, I do not expect that a full discussion period would have led to meaningful upward score revisions. While one reviewer viewed the work as marginally above the acceptance threshold, they explicitly indicated that rejection would also be reasonable. Reviewers who expressed concerns about novelty, evaluation completeness, and experimental rigor would likely have maintained their original scores. Overall, even under a full discussion scenario, the paper would most plausibly remain in the reject or borderline-reject range rather than converging toward acceptance.

---

### Decision · Program_Chairs · 2026-01-26

Reject